# Sex and Gender Identities Are Emergent Properties of Neural Complexity

**DOI:** 10.3390/bs15121599

**Published:** 2025-11-21

**Authors:** Simone Di Plinio, Olatz Etxebarria-Perez-De-Nanclares

**Affiliations:** 1Department of Neuroscience, Imaging, and Clinical Science, University of Chieti-Pescara, 66100 Chieti, Italy; 2Institute for Advanced Biomedical Technologies, University of Chieti-Pescara, 66100 Chieti, Italy; 3Faculty of Education and Sport, University of the Basque Country, 01006 Vitoria-Gasteiz, Spain; olatz.etxebarria@ehu.eus

**Keywords:** neural degeneracy, neurodiversity, evolutionary psychology, gender fluidity, identity diversity, behavioral plasticity

## Abstract

We investigate why the remarkable diversity of human identity, including gender fluidity, non-binary roles, and varied sexual orientations, is fundamentally rooted in the evolutionary and neurocognitive complexity of the human brain. Drawing upon interdisciplinary evidence from comparative biology, neuroimaging, anthropology, and social neuroscience, this paper explores how increased neural complexity across evolutionary trajectories supports behavioral plasticity and identity diversification. The concept of neural degeneracy, wherein different neural structures produce functionally similar outcomes, is central to understanding how individual and cultural diversity naturally emerges from the brain’s highly adaptable networks. By reviewing historical, prehistoric, and cross-species data, the paper demonstrates that identity diversity is neither recent nor culturally limited but has longstanding evolutionary and social foundations. Despite substantial scientific consensus on this inherent complexity, societal resistance persists, often driven by oversimplified and biologically reductionist interpretations of neuroscience. To counter these misunderstandings, the article introduces Complexity Neuroethics, a framework advocating the acknowledgment of diversity of identity expressions as an evolutionarily expected outcome of neurocognitive evolution. Ultimately, the review calls for a transformative dialogue between neuroscience and society, promoting policies, healthcare practices, and educational initiatives aligned with neuroscientific realities to foster more inclusive societies that embrace self-identity as an evolutionary and cognitive achievement.

## 1. Introduction

The human brain is widely recognized as the most complex biological system known, enabling an unparalleled range of cognitive, emotional, and social behaviors. Over the past two decades, neuroscience has increasingly focused on the idea that brain structural and functional complexity is not a static feature but an evolutionary achievement that allows for behavioral diversification, including the emergence of multidimensional identities. A key concept underlying this perspective is neural degeneracy, which refers to the capacity of structurally distinct neural systems to yield functionally equivalent outcomes ([37]; [98]). Degeneracy provides the neural substrate for both flexibility and individuality, allowing the same behavioral goal to be achieved through multiple neural configurations. This characteristic is particularly salient in humans, whose brains support complex repertoires of social roles, cognitive strategies, and self-conceptions ([120]; [115]). To avoid misunderstanding. In this article, the term *degeneracy* is used strictly in its technical, biological–neuroscientific sense, and thus to indicate “many-to-one” mappings in which structurally different neural assemblies can realize functionally equivalent outcomes. It does not carry any moral, social, or evaluative connotation and must not be read as suggesting that gender or sexual diversity is a “degenerate” or “lesser” form of behavior.

Recent theoretical models further underscore this idea. A recent review proposed a dual-pathway framework of neuroevolution that distinguishes between global trajectories (i.e., large-scale changes in brain architecture across species) and local trajectories, referring to species-specific expansion of behavioral repertoires ([27]). Within this model, human cognitive uniqueness is not attributed to isolated modules but to the emergent properties of highly integrated and interactive brain networks. Similarly, [16] ([16]) have shown that increased modularity and small-worldness of human connectomes are associated with flexible switching between network configurations, a core mechanism of adaptive behavior. The link between neural complexity and self-related processes has become especially relevant in recent years. Neuroimaging studies demonstrated that the multidimensional sense of self—encompassing the senses of self-identity and self-agency—arises from the coordinated integration and segregation of multiple brain systems, including the default mode network (DMN), salience network (SN), and frontoparietal network (FPN) ([29]). These networks are known to subserve internal mentation, interoceptive awareness, and executive control, respectively, and their dynamic interplay appears to scaffold the construction of a fluid and situationally adaptable sense of identity ([90]; [100]).

This functional plasticity, or what some scholars term “identity neuroplasticity” ([19]), implies that identity is not hard-wired but rather emergent from lifelong interactions between brain, body, and socio-cultural environments. Importantly, such plasticity is not a vulnerability or error but an evolutionary asset. From this vantage point, the diversity in gender identities and sexual orientations observed across human cultures can be understood as natural expressions of neurobiological variability, not as anomalies.

This article is a theory-oriented, narrative and integrative review aimed at bringing into dialogue bodies of literature that usually remain siloed (comparative ethology, archaeology, developmental biology, and network neuroscience) around a single explanatory axis: the link between neural complexity and identity diversification. Because our goal is conceptual integration rather than exhaustive coverage, we adopted a purposeful (rather than systematic) literature search and selection strategy. We first surveyed major journals and edited volumes in neuroscience, anthropology, archaeology, gender studies, and critical social science, using combinations of keywords related to neural complexity, degeneracy, sex/gender diversity, non-binary roles, queer archaeology, and identity. We then complemented these searches with backward- and forward-citation tracking from a set of seed articles and monographs that are widely cited or theoretically central in each domain. Inclusion criteria were: (a) peer-reviewed or otherwise widely recognized scholarly contributions; (b) direct relevance to at least one of our core constructs (neural complexity, identity diversity, gender/sexual variance, or biocultural plasticity); and (c) conceptual leverage for articulating links between brain organization, evolutionary trajectories, and social roles. We excluded purely technical reports without theoretical implications and non-scholarly commentaries that did not add new empirical or conceptual content. Within this purposeful framework, we deliberately included both inductive and deductive approaches, including more binary or essentialist accounts of sex/gender where they are influential, to avoid relying only on sources that straightforwardly support our central argument. This strategy does not eliminate selection bias, but it makes explicit the logic by which representative studies were identified and integrated in this narrative review.

Our purpose is to explore how the evolutionary expansion of brain complexity has facilitated the emergence of multidimensional identity spectra encompassing gender fluidity, non-binary roles, and diverse sexual orientations. Drawing on evidence from comparative biology, paleoanthropology, neuroimaging, and social neuroscience, we argue that such diversity is not only consistent with evolutionary theory, but in fact, an expected outcome of the human brain’s capacity for symbolic abstraction and behavioral flexibility. Crucially, we also examine how biologically reductionist frameworks, which seek to pathologize or marginalize non-conforming identities based on misinterpretations of neuroscience, fail to account for this inherent complexity. We aim to dismantle these frameworks by integrating data from multiple scientific domains, thereby promoting a more inclusive, evidence-based understanding of identity diversity.

## 2. Neuro-Behavioral Complexity Across Species

The emergence of gender and sexual diversity in humans must be understood within the broader evolutionary framework of increasing behavioral and neural complexity across species. Comparative neuroethology has shown that behavioral variance (including non-reproductive sexual behaviors, social bonding strategies, and role plasticity) is particularly prominent in species with more complex nervous systems and cooperative social structures ([25]; [72]). These findings challenge the view that binary sex roles are evolutionarily fixed, instead suggesting that flexibility and fluidity are often byproducts of neural sophistication.

One of the clearest examples comes from non-human primates. In wild chimpanzees, juveniles (especially females) have been observed carrying sticks in ways resembling rudimentary doll play; this sex-biased pattern parallels children’s play preferences ([67]). Complementing these field observations, experimental work in rhesus macaques shows that males preferentially interact with wheeled objects whereas females exhibit more variable preferences, mirroring human sex differences even in the absence of direct human socialization pressures ([55]). Importantly, these findings concern domain-specific sex differences in object play; in our framework they are distinct from self-identity plasticity, which we treat as a neurocognitive property within the broader sense of self.

This behavioral plasticity is echoed in other highly social mammals, particularly cetaceans. In bottlenose dolphins (Tursiops truncatus), male-male alliances are frequently reinforced through same-sex genital interactions, which seem to cement long-term coalitions critical for reproductive success and dominance hierarchies ([21]; [78]). These interactions suggest that non-heteronormative behaviors can be evolutionarily adaptive, especially in species where group cooperation determines access to resources and mates.

Importantly, such behaviors are not random or anomalous. They are evolutionarily stable strategies, widely documented across species. [8]’s ([8]) seminal work Biological Exuberance catalogued over 1500 species with documented cases of homosexual, bisexual, or transgender-like behaviors. More recent reviews have confirmed and extended these findings ([9]; [113]; [84]), noting that sexual behavior in animals serves multiple functions beyond reproduction: social bonding, conflict resolution, practice, play, and even exploration of social status.

This multifunctionality of sex becomes more pronounced in species with large brains relative to body size, often interpreted as a proxy for social and cognitive complexity ([35]). In these species, social dynamics are fluid, coalitions shift, and behavioral roles are negotiable, mirroring the non-essentialist logic that underpins human identity diversity. For instance, in Japanese macaques and rhesus monkeys, same-sex mounting is common and often serves to reinforce dominance or solidarity ([124]; [41]). These acts are not merely byproducts of captivity or overpopulation, as sometimes claimed, but appear in stable wild populations as well.

From an evolutionary perspective, these behaviors reflect the principle of exaptation—where traits evolved for one purpose are co-opted for another ([52]). In social mammals, neural circuits evolved for reproduction may become co-opted for social bonding and communication, especially in species where survival depends on long-term cooperation. This reframes non-heterosexual behaviors not as evolutionary “errors,” but as reproductive byproducts of social intelligence.

Additionally, these findings challenge simplistic Darwinian interpretations that equate fitness solely with reproductive success. As [108] ([108]) argues, sexual selection includes not just competition for mates but also negotiation, cooperation, and even alliance politics. In this light, diverse sexual behaviors, same-sex acts included, can serve as evolutionarily beneficial mechanisms to maintain in-group harmony, enhance group cohesion, or establish trust, all of which are crucial in species like humans.

To connect this more explicitly to human evolution: the expansion of the prefrontal cortex, increasing synaptic connectivity, and the elaboration of the social brain network ([48]) have enabled the Homo lineage to engage in symbolic thought, empathy, and social abstraction. These traits provide the cognitive infrastructure for a fluid and situationally modulated sense of self, capable of accommodating non-binary identities and diverse orientations ([2]; [75]).

Thus, far from being an exception to nature, the spectrum of human identity is a natural extension of neuroevolutionary trajectories observed in other intelligent species. Gender fluidity, same-sex attraction, and non-binary roles emerge not in spite of our biology, but because of the cognitive affordances it grants us. While non-reproductive behavioral diversity is evident across species, it is equally crucial to examine how human societies have historically recognized and institutionalized such diversity.

## 3. Historical and Cross-Cultural Evidence of Gender and Sexual Diversity

From a sociological and intersectional perspective, gender is best understood as a social construct, a historically contingent system of meaning that, in interaction with other identity categories such as race, class, and sexuality, both shapes individual self-definition and sustains social hierarchies. Social constructionism holds that reality and knowledge are formed through language, shared stories, history, and social interaction. It challenges the idea that knowledge is simply discovered, instead arguing that it is shaped by cultural and social processes. Individuals’ perspectives develop from their background and interactions, which over time create shared understandings of the self, others, and the world ([97]).

As articulated in multiracial feminist theory, gender and race function not only as sources of identity but also as principles of social organization that regulate access to power and resources ([14]). This interpretation aligns with our conceptual framework: while self-identity plasticity refers to neurocognitive adaptability within the sense of self, gender and sexual diversity operate at the socio-cultural level, where hierarchical norms that scholars refer to as heteronormativity, cisnormativity or gender-normative stereotypes, cultural scripts, and institutional structures mediate how identity expressions are enabled or constrained ([12]; [77]; [82]; [83]; [92]). Therefore, it is important to recognize that gender is a multifactorial concept. The Gender Spectrum framework identifies three components: *biological gender*, based on sex assigned at birth, external presentation referred to as *gender expression*, and *gender identity*, which is a reflection of the individual’s internal sense of self ([42]).

While comparative neurobiology reveals that behavioral diversity is biologically embedded across species, archaeology and anthropology confirm that gender and sexual diversity has also been socially recognized throughout human history. Contrary to the assumption that non-binary identities and same-sex relationships are recent cultural inventions or Western constructs, historical and prehistoric records suggest that such diversity has deep cross-cultural and diachronic roots. This section surveys well-documented cases attesting to gender role variance, cross-gender presentation, and early gender-affirming practices. In our framework, these are domain-specific phenomena (gender/sexual diversity) and are analytically distinct from self-identity plasticity, which we use to denote neurocognitive adaptability within the broader sense of self. Where examples derive from myth or literary sources, we treat them as cultural representations rather than empirical records.

### 3.1. Classical and Early Modern Attestations

Across classical sources and early modern records, we find domain-specific evidence of gender and sexual diversity alongside cultural representations of gender variance. In Greek myth, the figure of Tiresias, transformed first into a woman and later back into a man, illustrates a long-standing cultural imagination of cross-gender experience (Ovid, Metamorphoses III). These narratives are mythic representations, not empirical reports, yet they reveal that gender variance was thinkable within premodern frameworks. Scholarly readings emphasize that punitive interpretations of the metamorphosis are far from uniform across sources and epochs ([24]).

Moving from myth to Roman history, textual testimonies describe practices and roles that challenge binary expectations. Ancient historians report that Nero married the eunuch Sporus, reportedly castrated and presented publicly as a bride. The episode is preserved in Suetonius and Cassius Dio, Roman History, often read today as evidence of same-sex marriage rituals and gendered role-play at the imperial court ([1]). A century later, the emperor Elagabalus is portrayed by Cassius Dio as requesting physicians to create female anatomy by incision, an oft-cited but controversial passage that modern historians interpret cautiously due to rhetorical bias in the sources ([58]). Nevertheless, it indicates that ancient writers could imagine surgical alteration of sexed anatomy.

In early modern Europe, we move from literary testimony to documented life histories. The abbé François-Timoléon de Choisy wrote memoirs describing lifelong cross-dressing within elite French society, offering a window into the interplay of gender performance and social anxiety in seventeenth-century France. The Chevalier(e) d’Éon (1728–1810) left an unusually rich autobiographical and archival record; modern scholarship debates categories but recognizes d’Éon’s centrality to the European genealogy of gender variance. Building on such cases, sexologist Havelock Ellis coined the term “eonism” in the early twentieth century to distinguish cross-gender identification and presentation from homosexuality ([40]).

The nineteenth century also offers contested biographies such as James Barry, a military surgeon who lived and worked as a man and was revealed posthumously to have been assigned female at birth. Archival and curatorial accounts confirm the biographical core and discuss the historiographic debates surrounding contemporary gender terminology ([36]).

Finally, the early twentieth century marks the transition to experimental gender-affirming surgeries in Europe. At the Berlin Institute for Sexual Science, Dora Richter is documented as one of the first trans women to undergo orchiectomy (1922) and later vaginoplasty (1931); Lili Elbe underwent a series of highly experimental procedures between 1930 and 1931 ([68]). While the original medical records were destroyed, multiple secondary reconstructions converge on the chronology and context of these pioneering interventions.

To note, mythic narratives reflect symbolic and didactic aims rather than empirical documentation, while early modern biographies were often mediated by satire, censorship, or confessional bias. Consequently, our use of these sources is illustrative rather than diagnostic: they exemplify the social recognizability of gender and sexual diversity across time, not direct evidence of neurocognitive mechanisms underlying self-identity plasticity. However, these materials represent domain-specific evidence of gender and sexual diversity across periods and cultures. Cross-gender roles, gender-nonconforming presentation, and gender-affirming practices are neither modern inventions nor exclusively Western phenomena, even though their categories and meanings vary with genre, source bias, and medical-legal context.

### 3.2. Indigenous and Traditional Societies: Beyond the Gender Binary

Many precolonial societies around the world recognized and institutionalized gender roles that transcended the male/female dichotomy. One of the most widely studied examples is the Two-Spirit identity among Indigenous peoples of North America. The term, coined in the 1990s as a translation of traditional concepts, refers to biologically male or female individuals who fulfilled mixed-gender roles in their communities. Anthropological records document over 130 North American tribes that acknowledged Two-Spirit people as holding ceremonial, spiritual, and social functions beyond conventional gender norms ([107]; [61]; [33]).

These individuals often served as healers, storytellers, matchmakers, adoptive parents, or more generally roles that were socially valuable and spiritually revered. Importantly, their status was not considered marginal or deviant before the onset of colonization. Rather, colonial interventions and missionary frameworks introduced heteronormative hierarchies, suppressing fluid gender systems and stigmatizing Two-Spirit individuals ([87]). As a result, many Indigenous communities today are working to revitalize and reclaim these identities as a form of cultural resilience and resistance ([128]).

Similar phenomena are found in other parts of the world. In South Asia, the Hijra community, recognized legally in India, Pakistan, and Bangladesh, has existed for centuries as a distinct social category, composed of transgender, intersex, or third-gender individuals who traditionally held ceremonial roles ([102]). In the Samoan context, the fa’afafine, people assigned male at birth who embody both masculine and feminine traits, are an accepted and often integrated part of society, not considered anomalous but part of the gender continuum ([109]; [125]). Comparable roles, such as the waria in Indonesia and the bakla in the Philippines, further demonstrate that non-binary and fluid gender identities have long been part of cultural fabric outside the Global North.

Even in pre-modern Europe, alternative gender roles occasionally emerged in ritualistic and spiritual contexts. The Galli, self-castrated priests of Cybele in ancient Rome, performed sacred functions while adopting feminine dress and behavior ([106]). Among the Scythians, Herodotus described the enarees, gender-nonconforming shamans believed to possess special divinatory powers ([56]). These historical cases challenge the notion that the gender binary is a universal or timeless truth; rather, they highlight how cultural frameworks shape, regulate, and sometimes celebrate identity diversity.

### 3.3. Prehistoric Burials and the Archaeology of Non-Binary Roles

Prehistoric evidence further supports the presence of non-binary social categories in early human communities. While interpreting ancient identity is methodologically challenging, archaeological methods have advanced to detect gender variation in funerary contexts through a combination of osteological analysis, burial positioning, and grave goods ([69]). A prominent example comes from Central Europe’s Early Bronze Age burials, where individuals with male skeletal traits have been found interred with items culturally associated with female roles (and vice versa). A recent systematic analysis of burial data across time periods and regions proposed moving beyond the assumption that such cases represent anomalies or errors. Instead, these may reflect institutionalized third-gender categories or accepted gender variance within those societies ([93]).

Notably, the famous “Red Lady of Paviland”, a Paleolithic burial in Wales long assumed to be female because of ochre pigmentation and rich ornaments, has been conclusively reidentified as a biologically male individual, prompting systematic reevaluations of gendered assumptions in mortuary archaeology ([4]; [60]). Parallel debates at major Eurasian sites show that gender patterning in deathways was often subtler than once portrayed: at Çatalhöyük, multi-strand bioarcheological work (stable isotopes, osteology, imagery, and burial context) indicates limited sex-based differentiation and a stronger role for age and household in structuring mortuary practice, complicating binary readings ([96]). Comparable discussions in the Danube Gorges (including Lepenski Vir) likewise emphasize context and life stage over strict gender dichotomies in burial treatment ([101]; [105]).

New osteological and isotopic syntheses also indicate that women directly participated in hunting during the Late Pleistocene and early Holocene. A key example is the 9000-year-old female at Wilamaya Patjxa (Peru), interred with a complete big-game hunting toolkit, one of the clearest archaeological demonstrations that women in early forager societies pursued large game ([54]). Ethnographic meta-analysis further shows that between 30% and 50% of big-game hunters in early human groups may have been female ([54]; [127]; [118]), and that where game size is specified, women pursue large game in roughly one-third of cases, undercutting rigid “male hunters/female gatherers” models and aligning with queer-archaeological frameworks that foreground variability and overlap in labor roles ([6]). These findings received renewed attention in 2024–2025 following broader syntheses of prehistoric burials and subsistence practices, with multiple independent teams confirming similar evidence in Asia and Europe ([86]). Collectively, these results undermine the long-held binary narrative of “male hunters and female gatherers” and reinforce the view that gendered labor roles in prehistoric societies were fluid, overlapping, and context-dependent, much like the non-binary burial patterns identified through queer archaeological frameworks.

Recent anthropological syntheses further expand the record of gender and labor diversity in deep time. A paired set of articles in American Anthropologist reviews convergent archaeological and physiological evidence that women routinely hunted game across foraging contexts, undermining the binary “Man the Hunter” narrative and situating subsistence roles as flexible, overlapping, and context-dependent ([71]; [91]). Complementing prehistoric cases, comparative ethnography documents stable, socially recognized variations in gender categories and roles across societies ([89]).

Together, these lines of evidence reinforce our core claim: identity and role diversity are longstanding human regularities rather than modern inventions. These converging strands of evidence urge archaeologists to deconstruct the binary lens through which past societies have been interpreted. The recent shift toward queer archaeology seeks to expose the heteronormative biases that have long influenced assumptions about past identities and to explore how social roles might have been symbolically and structurally more flexible than previously assumed ([30]; [11]).

### 3.4. Symbolization, Identity, and Cognitive Pluralism

The historical prevalence of non-binary roles is closely linked to the evolution of symbolic cognition, which is our species’ capacity to represent and communicate abstract concepts, including the self. From figurines with ambiguous gender traits (e.g., the “Venus” of Willendorf or the so-called “shamanic figures” from Gravettian cultures) to ritual objects interpreted as mediating between masculinity and femininity, symbolic artifacts provide insight into how prehistoric humans conceptualized identity ([20]; [51]).

Such pluralization of identity is congruent with the neuroevolutionary view presented in previous sections: as brains became more capable of abstract thought and recursive self-representation, identity moved from a fixed biological assignment to a negotiable, contextual, and culturally embedded construct. This explains why, in so many diverse settings and epochs, societies have created language, roles, and rituals to strictly tied to identities extended beyond binary norms.

## 4. Evolutionary Trajectories and Identity Diversification

### 4.1. From Evolutionary Complexity to Neural Degenerate Code

The integration of evolutionary, archaeological, and neurocognitive evidence points to a compelling conclusion: the human brain’s complexity is a driving force behind the diversification of identity and behavior. As hominin brains evolved expanding in volume, connectivity, and functional specialization, they enabled not only enhanced problem-solving and social coordination, but also the emergence of fluid, context-sensitive self-representations ([116]; [35]).

This transition marked a shift from rigid, stimulus-driven responses to symbolically mediated behavior, allowing humans to construct roles, affiliations, and identities that are not bound to biological imperatives ([119]). In this framework, identity (encompassing gender, sexual orientation, and self-perception) is best understood not as a fixed trait but as a neurocognitively plastic phenomenon, shaped by ongoing interactions between neural substrates, personal experiences, and sociocultural feedback loops.

A central concept in this discussion is functional degeneracy, i.e., the idea that multiple neural pathways can support similar outcomes. This principle not only explains resilience and adaptability in cognition but also provides a neural basis for individual uniqueness in identity formation. [37] ([37]) describe degeneracy as a feature, not a flaw, of biological systems: it allows for variability within constraints, a property essential for the emergence of complex behavioral phenotypes, including those related to the self. There are many other influential neuroscientists discussing and highlighting the importance of considering the quasi-chaotic, degenerate features of the brain, both implicitly and explicitly. For example, [120] ([120]) and [47] ([47]) extend this idea by showing how the brain’s architecture is optimized not for rigid behavior but for metastability: the ability to switch flexibly between states while maintaining coherence. This neurodynamic flexibility underpins self-continuity despite variability, allowing identity to remain stable across time even as it adapts to new experiences and social roles ([28]).

Work in evolutionary biology and human behavioral ecology suggests that sexual selection in humans increasingly operates via social performance, signaling, and culture-mediated criteria rather than fixed dimorphisms, thereby favoring cognitive, behavioral, and identity plasticity. Recent syntheses argue that human mating and status displays are deeply scaffolded by symbolic performance and social learning ([99]), and that the evolution of human sexuality reflects broad cooperative and cultural dynamics beyond binary templates ([74]; [49]). This framing aligns with our degeneracy account: selection of flexible, performative traits promotes many viable routes (e.g., neural and social) by which individuals realize identities.

### 4.2. Endocrine and Neurobiological Modulators of Identity

Alongside network degeneracy, endocrine signals contribute probabilistically to identity-relevant development. In the organizational–activational framework, transient surges of gonadal steroids during sensitive periods bias neural circuit formation (organizational effects), while later hormonal milieus modulate the expression of these circuits (activational effects) ([110]; [81]). In humans, convergent evidence suggests that prenatal androgen exposure is associated with sex-typed interests and, for some individuals, later sexual orientation ([57]). Reviews of sexual differentiation of the human brain similarly indicate that steroid hormones shape structural and functional phenotypes across development, again with broad variability and context dependence ([10]). Importantly, such influences are probabilistic, not categorical: they interact with genotype, experience, and enculturation, aligning with our view that self-identity plasticity emerges from the coupling of biological potentials with socio-cultural scaffolds.

Evidence from hormone interventions reinforces this plastic view. In transgender men, testosterone therapy is associated with measurable macro- and micro-structural brain changes (e.g., cortical volume, subcortical and hypothalamic measures), alongside functional adaptations detectable with multimodal magnetic resonance imaging ([17]; [131]; [70]). Meta- and mega-analytic work likewise reports patterned, but non-deterministic differences between transgender and cisgender groups, further supporting a model in which endocrine context modulates, rather than fixes, identity-relevant neural dynamics ([88]; [46]). These findings dovetail with the brain-as-mosaic perspective articulated below: hormones shape distributions and tendencies, but do not yield discrete, essentialized brain types.

In the context of identity diversity, this means that diverse trajectories of gender expression, sexual orientation, and role adoption are not neurobiological deviations but emergent properties of a complex, self-organizing system. As the number of potential neural configurations increases with brain complexity, so too does the number of viable identity configurations, each coherent within its own neural and social context.

Contemporary social neuroendocrinology emphasizes that hormone–behavior relations are context-dependent and bi-directional. Sexual Configurations Theory integrates gender/sex, partner variables, and eroticism to model individual variability without presuming dimorphism ([122]). Experimental work demonstrates that enacting competition rather than “masculinity” per se can up-regulate testosterone, evidencing gender-hormone pathways shaped by social scripts ([123]). These insights converge with cautions about interpreting “sex differences” outside their contexts ([76]) and with updated meta-scientific arguments to take the null hypothesis seriously in human brain (and sex/gender) research ([38]). Finally, scholarship on testosterone’s sociohistorical meanings shows why simplistic hormonal essentialism is empirically and conceptually untenable ([65]).

### 4.3. Network Dynamics, Brain Mosaic, and Biocultural Plasticity

Modern network neuroscience further elucidates the mechanisms through which identity diversification emerges. Studies have identified the default mode network (DMN) as a central hub in self-referential thinking, including autobiographical memory, perspective-taking, and future simulation ([15]; [7]). The salience network (SN) and frontoparietal control network (FPN) modulate attentional shifts between internal and external states, allowing dynamic reconfiguration of the self in response to social and environmental demands ([112]; [121]). In [29] ([29]), task-based fMRI revealed that multidimensional self-representations, including perceiving owns’ sensory features and actions, recruit overlapping yet partially dissociable networks, suggesting that the self is a composite, distributed phenomenon. Their findings support the view that selfhood is not anchored in a single brain region but arises from inter-network coordination, susceptible to modulation through learning, social context, and introspection.

This distributed architecture explains why identity is both robust and flexible. For example, individuals undergoing gender transition often report changes in self-perception that are reflected in neuroimaging findings such as altered connectivity in the DMN and insula, regions implicated in self-location and body awareness ([79]; [17]). These changes are not “abnormalities” but indicators of the brain’s capacity to update and recalibrate identity representations considering embodied experience and intentional self-construction.

Perhaps one of the most direct neuroscientific refutations of essentialist notions of gender is the so-called concept of “brain mosaic”. In a landmark study, [62] ([62]) analyzed over 1400 brain scans and found that individual brains do not conform to a binary classification but instead exhibit mosaics of features statistically more common in either males or females. Subsequent meta-analyses have confirmed that while sex differences in brain structure and function exist at the population level, they do not cluster into discrete “male” and “female” types ([104]; [39]). A recent commentary further updates this position, recommending explicit consideration of the null when evaluating human brain sex/gender differences ([38]).

This undermines the idea that gender identity or sexual orientation can be reliably inferred from brain anatomy or function. Instead, it suggests that neurobiological variability is the norm, and that identity and behavioral diversity (e.g., gender-nonconformity, sexual fluidity, queerness) is not aberrant but within the expected range of human variation.

Finally, emerging interdisciplinary frameworks such as neuroconstructivism and neuroanthropology emphasize that identity results from biocultural entanglements: neural circuits are shaped by both evolution and enculturation ([63]; [73]). This perspective resonates with Vygotsky’s seminal notion that cognitive and affective development are mediated by social interaction and cultural tools, through which individuals internalize and transform shared meanings ([126]). Children’s gender identities, for example, are formed through interactions with caregivers, language, toys, clothing, and social scripts, which are factors that dynamically interact with innate predispositions and brain plasticity ([44]; [80]). This view dissolves the binary opposition between “nature” and “nurture,” recognizing instead that identity is co-constructed by neural potential and cultural scaffolding. It is precisely the plastic, self-reflective nature of the human brain that enables such diversity to flourish.

In sum, identity diversification is not a disruption of neurobiological norms but a manifestation of evolutionary success. Diversity is a testament to our brain’s capacity to transcend rigid scripts and co-author narratives of selfhood that are richly varied, socially embedded, and biologically supported.

## 5. Cultural Resistance to Neurobehavioral Diversity

Despite growing scientific consensus on the multidimensional and biologically grounded nature of gender and sexual diversity, cultural, political, and institutional resistance to non-conforming identities remains widespread. These forms of resistance are often rooted in reductionist misinterpretations of neuroscience and biology, invoking naturalistic fallacies and pseudo-scientific arguments to justify exclusion, pathologization, and moral condemnation.

### 5.1. The Persistence of Biological Essentialism

A primary mechanism of cultural resistance is biological essentialism, i.e., the belief that identity, behavior, and social roles are determined solely or primarily by biological sex. This viewpoint, long discredited in contemporary biology and neuroscience, persists in public discourse and some policy debates, where it is used to deny the legitimacy of transgender, intersex, and non-binary people ([64]; [44]). Essentialist arguments frequently rely on simplistic interpretations of dimorphic traits, such as chromosomal patterns (XX vs. XY), genital anatomy, or hormone levels, to draw rigid lines around identity categories. However, as research in developmental biology and endocrinology has shown, sex is not a binary variable, but a multidimensional construct involving chromosomal, gonadal, hormonal, anatomical, and neurological parameters, many of which do not align neatly within binary frameworks ([3]; [13]; [31]). The existence of intersex conditions—estimated to occur in approximately 1.7% of the population ([43])—further underscores the limitations of binary sex categories.

In neuroscience, similar distortions occur through neurosexist narratives ([45]), which exaggerate or misrepresent average differences between male and female brains to assert innate cognitive or emotional disparities. These claims often ignore overlapping distributions, environmental plasticity, and the lack of predictive validity at the individual level ([104]; [39]). For instance, while men and women may show statistically significant group-level differences in brain volume or connectivity patterns, these differences are minor and do not support categorical distinctions ([62]; [66]). Moreover, essentialist logic is frequently applied inconsistently: for example, critics demand “proof” that transgender individuals have the brain structure of the “opposite sex”, yet if such differences are demonstrated (e.g., via altered insular or cingulate connectivity), they are used to pathologize trans identity as a neurodevelopmental anomaly ([79]; [17]). This double-bind logic reflects an ideological bias, not scientific reasoning.

Legal governance provides a parallel caution: categories of “sex” in policy are not passive reflections of biology but state effects. In other words, categories can be thought of as administrative classifications that vary by domain (ID documents, prisons, sport, healthcare) and purpose ([23]). Recognizing this variability helps align neuroscientific nuance with rights-based frameworks and avoids exporting biological reductionism into law. To avoid re-inscribing essentialism, recent work proposes sex contextualism: treating “sex” not as a causal variable but as a classification composite whose relevant dimensions (e.g., gonadal hormones, gametogenesis, receptor expression, morphology) must be specified for the research question at hand ([103]; [94]). Such framework (advanced by the GenderSci Lab) offers concrete practices for operationalization, analysis, and reporting that increase rigor and reduce misinterpretation in basic and translational research.

### 5.2. Pathologization and the Legacy of Psychiatric Stigma

The pathologization of non-normative identities has a long and troubling history. Homosexuality was classified as a mental disorder in the Diagnostic and Statistical Manual of Mental Disorders (DSM) until 1973, and transgender identity was listed as “Gender Identity Disorder” in both the DSM-IV and the International Classification of Diseases (ICD-10). The DSM-5 ([5]) reframed it as “Gender Dysphoria,” shifting the focus from identity itself to the distress that may accompany incongruence between experienced gender and assigned sex. Subsequently, the World Health Organization adopted a similar reconceptualization: in the ICD-11 ([129]), the diagnosis was renamed “Gender Incongruence” and moved from the chapter on mental and behavioral disorders to that of conditions related to sexual health, thereby marking a decisive step in depathologizing gender diversity.

These classifications were not based on empirical evidence of dysfunction but on sociocultural discomfort and normative assumptions. As scholars such as [22] ([22]) and [32] ([32]) have shown, diagnostic frameworks often reflect societal biases more than scientific objectivity. Today, major health organizations including the American Psychological Association, the American Psychiatric Association (APA), and the World Health Organization (WHO) affirm that gender diversity is not pathological, and that conversion therapies aimed at changing sexual orientation or gender identity are unethical and harmful ([5]; [129]). Nevertheless, residual medicalization persists, particularly in legal and institutional domains. Transgender people are often required to obtain psychiatric evaluations, hormone therapy, or surgeries as prerequisites for legal gender recognition, a practice criticized for reinforcing the idea that trans identity requires “proof” of biological legitimacy ([114]). Such requirements perpetuate the notion that diverse identities must be scientifically justified to be respected, rather than accepted as valid on their own terms.

### 5.3. The “Against Nature” Fallacy, a Selective Use of Evolution

Another common resistance strategy involves the invocation of nature as a moral arbiter. Critics of LGBTQ+ rights often argue that non-heterosexual or non-cisgender identities are “unnatural” and therefore illegitimate. This appeal to nature, however, is not only philosophically flawed ([85]), but also empirically false.

As reviewed in Section 2, same-sex sexual behavior and gender fluidity have been documented in hundreds of species across taxa ([8]; [9]; [84]). These behaviors are often functional within social systems, contributing to cooperation, alliance-building, and stress reduction ([25]). Moreover, anthropological and historical evidence shows that gender-diverse roles have long existed in human societies and were often revered ([107]; [102]). The claim that such identities are “modern inventions” or “Western ideologies” ignores the cross-cultural and historical ubiquity of neurobehavioral diversity. Yet, “naturalness” is invoked selectively. Many widely accepted human practices (such as monogamy, contraception, abstinence, or literacy) are not “natural” in the sense of being common across non-human animals, yet they are not condemned. This inconsistency reveals that appeals to nature are often rhetorical devices, deployed to uphold heteronormative and cisnormative ideologies, not coherent scientific arguments.

### 5.4. Resistance as Reaction to Complexity and Its Implications

Cultural resistance to identity diversity can be understood, in part, as a defensive reaction against cognitive and social complexity. As argued by scholars in critical psychology and queer theory, binary thinking offers epistemological comfort: it simplifies identity categories, social roles, and moral judgments ([18]; [111]). The fluidity and spectrum-based understanding of identity emerging from neuroscience and anthropology challenges this simplicity, requiring individuals and institutions to grapple with ambiguity, multiplicity, and change.

This epistemic discomfort fuels backlash, particularly in socio-political contexts marked by populism, nationalism, or religious fundamentalism, where “traditional values” are framed as under siege. In these discourses, gender and sexual diversity are not merely misunderstood: they are actively framed as threats to social order, morality, or the “natural” family ([95]; [53]). Yet, if one accepts the premise that the human brain is a generator of complexity capable of supporting flexible, non-binary identity formation, then such diversity must be seen not as moral deviation but as evidence and consequence of neurocognitive potential. To reject this is not only to ignore science, but to suppress the very human capacity for meaning-making, empathy, and symbolic selfhood.

## 6. Conclusions

The central perspective advanced in this article is that identity fluidity and diversity are not deviations from human biology but evolutionarily grounded outcomes of the brain’s intrinsic complexity. Throughout evolutionary history, increased neural complexity has supported richer and more flexible behavioral repertoires. In humans, this culminates in cognitive capacities that transcend rigid biological roles, enabling gender fluidity, diverse sexual orientations, and non-binary identities. Such expressions are not anomalies but expected manifestations of our neurobiological architecture. What we are therefore putting forward is not only a synthesis of existing evidence, but a testable, inclusive theory of identity and gender fluidity as emergent properties of biologically complex, socially embedded nervous systems. Such a theory can be operationalized, for example, by examining whether higher-order network flexibility and cultural affordances statistically predict broader identity repertoires across populations, and it can be falsified if future data show stable, low-complexity neurocognitive architectures coexisting with high identity diversification. As a narrative, non-systematic review, this work is necessarily exposed to selection bias. Our aim was not to provide an exhaustive or quantitatively balanced survey of all available evidence, but to develop a theory-driven framework that integrates representative findings from neuroscience, anthropology, archaeology, and related fields. The literature search and selection strategy described in the Introduction is intended to make our choices transparent and to mitigate the risk that we prioritize studies aligned with our central thesis. Future systematic or scoping reviews could build on this conceptual synthesis by formally assessing the breadth, strength, and possible gaps in the empirical support for each component of the proposed framework

Neuroscientific findings including neural degeneracy, distributed network functionality, and the mosaic structure of sex-related brain features demonstrate that human identity cannot be reduced to essentialist or dimorphic frameworks ([62]; [104]; [37]; [28]). Identity emerges as a dynamic integration of biology, experience, and culture, supported by the brain’s evolved capacity for plasticity and interaction ([29]; [44]; [119]; [50]). These findings challenge the legitimacy of any binary, reductive categorization of identity and call for a paradigm that aligns with scientific complexity.

Despite this, neuroscientific evidence is frequently misrepresented in public and political discourse. Popular narratives often cherry-pick or distort research to reinforce stereotypes, promote exclusionary ideologies, or pathologize identity diversity. When used as a rhetorical weapon, neuroscience risks being stripped of its epistemic and ethical integrity. This misuse not only distorts empirical findings but undermines the core responsibility of science: to foster understanding, accuracy, and dignity.

To address this, we propose an integrative framework: Complexity Neuroethics. Grounded in the principles discussed throughout this review and previous work on neuroethical responsibility ([59]; [130]; [26]), this framework asserts that:Complexity is the evolutionarily expected pattern. Neural, cognitive, and behavioral complexity is a recurrent, resilient outcome of human neuroevolution; attempts to compress identity and roles into rigid binaries yield at most local, temporary simplifications, after which variation re-emerges. Contemporary evolutionary and anthropological syntheses show that human sexual selection and socioecology favor flexible, performative traits and role diversity including cross-culturally and in deep time (e.g., hunting as a non-binary, shared activity), consistent with our degeneracy account. ([99]; [71]; [74]; [49]; [27]).Identity is emergent and dynamic. It arises from continuous brain–body–culture loops and from context-sensitive psychoneuroendocrine dynamics: social scripts and enacted roles can modulate hormones just as hormones can modulate behavior, and individual sexual configurations are multidimensional rather than dimorphic. This supports treating identity diversity as expected, fluid, and multidimensional. ([122]; [123]; [76]).Simplification equals distortion. Reducing neuroscientific findings to binary categories introduces analytic error: large, recent syntheses emphasize small, overlapping, and often non-replicable brain differences across sexes and recommend explicitly considering the null hypothesis of no difference as a default starting point. Treat “sex” not as a causal variable but as a composite classification whose relevant biological dimensions must be specified for the question at hand. ([38]; [94]).Ethical responsibility of communicators. Scientists, clinicians, educators, journalists, and policymakers should resist oversimplification by adopting contextualist operationalizations of sex-related variables, reporting limitations and uncertainty, and avoiding the export of biological reductionism into law and policy. Legal categories of “sex” are administrative instruments that vary by domain and purpose, which is why translation from lab to policy requires special care and clear scope conditions. ([23]; [34]; [94]).

A key implication of our arguments is that identity diversity is not caused by culture. What culture does is to format, name, authorize, follow, or suppress identity configurations that are already made possible by behavioral & neural complexity and degenerate coding. The recurrent historical appearance of non-binary, third-gender, and gender-crossing roles in distant societies ([107]; [102]; [117]) suggests that, once a sufficiently complex brain meets symbolization and social exchange, some form of identity pluralization emerges. Cultures may narrow or widen the channel through law, religion, kinship, and medicine, but the underlying pressure for variation does not disappear, which is why new categories keep being coined even in already saturated classificatory systems. Put differently: human identity systems are generative. Even in social contexts that strive for categorical closure, individuals and groups reintroduce nuance, hybridity, or in-betweenness. This can be seen as the socio-cultural expression of the same principle we observe in neural systems: multiple realizability under constraint. Different people, in different epochs, will find or invent identity positions that better match their embodied and relational experience.

Implementing Complexity Neuroethics means aligning policy, education, healthcare, and public discourse with the neuroscientific reality of human variation. In education, promoting neuroscientific literacy across all levels can help dismantle reductive views on gender and identity and foster critical understanding of human diversity. In healthcare, clinicians must abandon pathologizing frameworks in favor of affirmative practices that support identity diversity as a natural aspect of cognitive functioning. In media and science communication, ethical standards must prioritize accuracy over sensationalism, conveying the complexity of the brain and identity responsibly. In legislation, policy-makers should ground laws on gender recognition, anti-discrimination, and healthcare access in frameworks that reflect neuroscientific insights, protecting identity expression as a matter of human dignity and rights.

Ultimately, this article calls for a reframing of how neuroscience engages with society. Rather than being co-opted by reactionary forces seeking to simplify and distort, neuroscience must become an active agent of progress. By embracing complexity as both a scientific principle and an ethical imperative, neuroscience can contribute to dismantling prejudice, promoting inclusion, and cultivating a deeper understanding of what it means to be human.

Recognizing identity fluidity as an evolutionary and neurocognitive achievement has far-reaching implications. Theoretically, it compels a re-evaluation of essentialist assumptions across neuroscience, psychology, and anthropology, and calls for methodologies capable of capturing the dynamic processes underlying diverse identities. Practically, it informs clinical care, educational programs, and legal protections grounded in evidence rather than dogma. And ethically, it urges the scientific community to fulfil its broader societal role, not only as the custodian of knowledge, but as a force for empathy, inclusion, and human dignity.

## Data Availability

No new data were created or analyzed in this study. Data sharing is not applicable to this article.

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
