# Peer review of "Sex and Gender Identities Are Emergent Properties of Neural Complexity"

_behavsci, 2025, doi:10.3390/bs15121599_

Round 1
Reviewer 1 Report
Comments and Suggestions for Authors
I am in strong accord with the core premise and the structured argument of the manuscript “Identity and Gender Fluidity are Emergent Properties of 2 Neural Complexity.” My only two comments of any substance are a) that some of the sections could be expanded and fleshed out/bolstered more fully as the topic and central argument of the MS will be criticized by a not insignificant subset of scholars, and b) that the references/citations be augmented by inclusion of a larger array of the slew of recent (last 3-5 years) of work in sex biology (writ large) and gender. This is a potentially important paper that could end up being read and cited widely so, given the topic, it has a responsibility to be as up to date and comprehensive in recognizing and acknowledging the broad, and diverse, cross-disciplinary suite of scholars/work in this area.
Specifically, I would suggest that section 3.3 (Historical and cross-cultural evidence of gender and sexual diversity) could benefit from including the recent human bio and paleo-archeological work of Cara Ocobock and Sarah Lacy (their twin 2024 Amer. Anthropol. Articles ), and some of the historical and contemporary gender diversity work of Paisely Currah, Serena Nanda, and others. Section 4 could also nod to a bit more of the recent work on human cognitive and social diversity and its role in human evolution and in psychoneuroendocrine dynamics. The work of Sari van Anders and Donna Maney would be good to connect here and while the one nod to Lise Elliot is good, there is a set of recent work by her on neurobiology and gender that is also relevant (the 2019 book by Rebecca Jordan-Young and Katrin Karkazis is also worth having a look at). I think that some engagement with the work of Sarah Richardson and the GenderSci lab would also be beneficial. Recent books from 2024-25 by Richard Prum, Nathan Lents, Agustin Fuentes, Paisley Currah, and the just released (and open access) “Sex and Gender: Toward Transforming Scientific Practice” edited by L. Zachary DuBois, Anelis Kaiser Trujillo, Margaret M. McCarthy, have a lot of information, framing, and data/analyses that could do positive work in bolstering the argument in this MS.
In sum, I am very excited t see this MS com out, but would ask the authors to enhance the connections to the recent relevant literature in order to both bolster their case and to recognize and acknowledge the scholars doing this work.
Author Response
Dear Reviewer,
Thank you for your thoughtful and encouraging evaluation of our manuscript. We carefully revised the paper to address your suggestions as fully as possible, while avoiding redundancy with existing content.
We expanded Section 3.3 to integrate recent anthropological and cross-cultural research, including the paired 2024 American Anthropologist articles by Cara Ocobock and Sarah Lacy, and a comparative synthesis from Serena Nanda, to strengthen the historical and cross-cultural evidence base.
At the start of Section 3, we also added a concise sociological and intersectional framing that treats gender as a historically contingent social construct, clarifies how hetero-cisnormative structures and cultural scripts shape expression, and explicitly adopts a multilevel view connecting these dynamics to our neurocognitive framework; we also introduce the Gender Spectrum framework to distinguish biological sex assignment, gender expression, and gender identity.
We augmented Section 4 with current work on human cognitive and social diversity in evolution (e.g., Prum 2024; Lents 2025; Fuentes 2025) and with a focused paragraph on psychoneuroendocrine dynamics drawing on Sari van Anders (Sexual Configurations Theory; experimental evidence of social modulation of testosterone), Donna Maney (reporting standards for sex differences), Lise Eliot (2024 commentary emphasizing the null), and Jordan-Young & Karkazis (on testosterone).
We engaged more directly with GenderSci Lab scholarship by introducing sex contextualism and related methodological guidance in Section 5.1 (e.g., Richardson 2022; Pape et al., 2024), and we added a brief policy note referencing Paisley Currah to align scientific nuance with governance considerations.
We updated and broadened the conclusion section as well the reference list accordingly.
We believe these revisions substantively strengthen the manuscript’s empirical grounding and interdisciplinary reach, and we are grateful for your guidance in achieving this.
Reviewer 2 Report
Comments and Suggestions for Authors
There are a few improvements that I would suggest. Two terms might be controversial, I would develop a little on two aspects and, maybe present a recognition that you are proposing a new inclusive and integrative theory regarding sexual and gender diverse identities and expressions.
Terms. In "Neural Degeneracy" The degeneracy term may have two meanings: the one given scientifically to what it describes and one giving place to the idea that sexual diversity is a degenerated behavior. In my early years of homosexuality, that was used to discriminate and support homophobic behaviors by judging people, we were “degenerated”. I guess that is not the goal of the authors.
In the expression of “Complexity is normative”, it may seem contradictory to traditional speakers that homosexuals may claim to be the normality or the norm when still too many people think they are not part of normality. I understand the meaning that is proposed but this is again easy to slip from one meaning to the next if the term is used here. I would suggest using a term speaking of its permanency or of its recurrency despite any attempts to simplify (and oversimplify) the idea. It is somehow resistant to simplification and adaptative, so it remains complex even after alteration. Somehow the Complexity is the effective norm regarding brain and identities, not “normative”. To be discussed among authors, I guess.
For additional arguments. The method used to create this article is a little more oriented than a usual literature review. This is not a review with “objective” process as articles of current years are proposed: not a systematic review and not a scoping review or other typical review (they compete with sexual diversity for creating new categories it seems, haha). So I see that your methods can be better described, and limits assessed for the non exhaustive collection of articles. The goal was to sew together all these domains instead of having an exhaustive coverage of complexity. E.g.: this is finally an essay based on review of existing literature in several domains that are complementary but do not speak to each other very often. You are trying to cover all domains for what they can speak to in terms of complexity and flexibility vs sexual diversity; your originality is to show a possible theoretical continuum between these domains of science that is parallel to human rights and politics (avoiding using them as justification) and getting away from essentialism.
Finally I would propose to include in the conclusion some aspects that were a strong deduction of your text, per example: that diversity is not a cultural product but is culturally created as outcome and that quest for being different is always present (whatever the culture you are from, the need for some diversity is always present in some people, over centuries and epoch), the interaction of complexity and culture creates the new diversity or its conformity/non-conformity with existing groups and categories. If we would find a name for all letters of the alphabet in sexual diversity, we may still need to find new ones as this is the role of “neural degeneracy” to quest for something that is different than what is known. Also, I would recall that Sex and Gender flexibility modulates identities and expression according to the psychosocial context and that identities and expression will always evolve with psychosocial and cultural contexts.
My understanding is that you are proposing (at least trying to propose) a new coherent and inclusive theory that may support the observations and science around sexuality and gender diversities. It would need to be tested as concept and as explanation in the background. With this theory, we don’t need to feel as a radical activist or as a victim or as a guardian of traditional values to accept using that theory which needs now new validations in the field. i.e. being LGBTQ+ or 2S or any identity that satisfies us is the result of our variate quest for different identities and expression of it and being here and now in our cultural psychosocial context.
Author Response
We sincerely thank the reviewer for their thoughtful and constructive feedback. We have carefully revised the manuscript to clarify terminology, improve transparency, and make the theoretical contribution more explicit.
We fully acknowledge the possible ambiguity of the term outside its scientific context. We have clarified that degeneracy is employed strictly in its biological and neuroscientific meaning, with no evaluative or moral connotation. We explicitly state this in the Introduction to prevent any misinterpretation.
Following the reviewer’s excellent suggestion, we reformulated this expression as “complexity is the evolutionarily expected pattern.” This wording preserves the intended idea of recurrence and resilience while avoiding potential misreadings related to social normality.
On the methodological nature of the paper, we clarified in the Introduction that this work is a theory-oriented, narrative and integrative review, rather than a systematic or scoping review. We describe the purpose and limitations of our approach to align with the reviewer’s observation that the goal was to connect complementary domains rather than to achieve exhaustive coverage.
On the theoretical scope and conclusions, now we explicitly state that the manuscript advances a testable and inclusive theoretical framework linking neural complexity to gender and sexual diversity. The Conclusions now emphasize that diversity is not a cultural product but a biologically grounded and culturally shaped phenomenon, continuously regenerated through the interaction between complexity and context. We also expanded the discussion of how sex and gender flexibility modulate identities in different psychosocial environments.
We are grateful for the reviewer’s insightful comments, which have significantly improved the clarity and inclusiveness of the manuscript.